

**The First Road Surface Type Dataset for 50 African**
**Countries and Regions**
Zixian Liu[1], Qi Zhou[1]*, Fayong Zhang[1]*, Prosper Basommi Laari[2]
*1. School of Geography and Information Engineering, China University of*
*Geosciences, Wuhan, People's Republic of China*
*2. Department of Environment and Resource Studies, Simon Diedong Dombo*
*University of Business and Integrated Development Studies, Wa, Ghana*
*Corresponding author: Qi Zhou (zhouqi@cug.edu.cn); Fayong Zhang
(zhangfayong@cug.edu.cn)





**Abstract**
Road surface types not only influence the accessibility of road networks and socio-
economic development but also serve as a critical data source for evaluating United
Nations Sustainable Development Goal (SDG) 9.1. Existing research indicates that
Africa generally have a low road paved rate, limiting local socio-economic
development. Although the International Road Federation (IRF) provides statistical
data on paved road length and road paved rates for certain African countries, this data
neither covers all African country nor specifies the surface type of individual roads,
making it challenging to offer decision-making support for improving Africa's road
infrastructure. To fill this gap, this study developed the first dataset for 50 African
countries and regions, incorporating the surface type of every road. This was achieved
using multi-source geospatial data and a tabular deep learning model. The core
methodology involved designing 16 proxy indicators across three dimensions—derived
from five open geospatial datasets (OSM road data, GDP data, population distribution
data, building height data, and land cover data)—to infer road surface types across
Africa. Key findings include: The accuracy of the African road surface type dataset
ranges from 77% to 96%, with F1 scores between 0.76 and 0.96. Total road length,
paved road length, and road paved rates calculated from this dataset show high
correlation (correlation coefficients: $0.69-0.94$) with corresponding IRF statistics.
Notably, the road paved rate also exhibits strong correlation with GNI per capita and
HDI (correlation coefficients: $0.80-0.83$), validating the reliability of the dataset.
Spatial analysis of African road paved rates at national, provincial, and county scales





revealed an average paved rate of only 17.4% across the 50 countries and regions. A
distinct "higher in the north and south, lower in the central region" pattern emerged, the
average paved rate north of the Sahara is approximately three times that of Sub-Saharan
(excluding South Africa). The African road surface type dataset developed in this study
not only provides data support for enhancing road infrastructure and evaluating SDG
9.1 progress in Africa but may also facilitate research on how road surface types impact
road safety, energy consumption, ecological environments, and socio-economic
development.
**Keywords:** Road surface type; multi-source geospatial data; SDG 9; Africa
**1.  Introduction**

Road surface types (such as paved and unpaved roads) not only affect vehicle

driving safety and energy consumption but also impact road accessibility and socio-
economic development (Anyanwu et al., 2009; Shtayat et al., 2020; Sha, 2021; Styer J
et al., 2024; Chen et al., 2025). Generally, paved roads have a sturdy structure and are
resistant to erosion, allowing them to be passable all-season, while unpaved roads may
be affected by natural factors such as rain and snow, making them typically difficult to
pass all-season. The proportion of the rural population living within 2 kilometers of all-
season road has also been adopted by the World Bank as an important indicator for
evaluating road infrastructure, and this indicator was incorporated by the United
Nations into the Sustainable Development Goal (SDG) 9.1 in 2017. Road surface type
data are considered one of the key data sources for assessing SDG 9.1.



Existing studies indicate that the road paved rate in African countries is highly
positively correlated with national poverty rates, and in some regions, the lack of all-
season passable roads has led to significantly increased transportation costs (Anyanwu
et al., 2009; Abdulkadr et al., 2022). Particularly in Sub-Saharan, more than 70% of
roads remain unpaved (Greening et al., 2010); In Nigeria, for example, over 30 million
rural residents have long been unable to access road transportation services. In these
countries and regions, the lag in transportation infrastructure has become one of the
main bottlenecks restricting socio-economic development (Li et al., 2022). To address
these challenges, the World Bank, the International Automobile Federation (FIA), and
the International Transport Forum (ITF) signed a Memorandum of Understanding
(MoU) in 2018, aiming to strengthen infrastructure construction in Africa over the next
fifty years (World Bank, 2018). The Agenda 2063: The Africa We Want, participated in
by multiple African countries, also sets goals to improve residents' quality of life and
enhance infrastructure in African nations (African Union Commission, 2018).
Therefore, high-quality road surface type data for Africa are of great significance for
improving local transportation infrastructure and promoting socio-economic
development.
However, the currently available, globally open road surface type data are primarily
statistical data, and most analyses of road surface types are also based on such statistics.
For example, the International Road Federation (IRF) provides statistical data related
to road surface types, such as paved road length and road paved rate (Turner, 2015; CIA,
2025). Greening et al. (2010) found, based on IRF and other road statistics, that in Sub-



Saharan, the proportion of "all-season road" (e.g., paved roads) does not exceed 30%.
Kresnanto (2019) used statistical paved road length data from Badan Pusat Statistik
Indonesia (BPS Indonesia) to analyze the relationship between road paved rates and
vehicle ownership in Indonesia from 1957 to 2016. Patrick et al. (2022) conducted a
survey to estimate the road paved rate in rural areas of Sub-Saharan. However, analyses
of road surface types based on statistical data have many limitations. On the one hand,
existing statistical data on road surface types do not cover all countries; for example, in
2020, IRF only provided statistics on paved road lengths for 19 African countries, and
some countries still face issues with untimely data updates (Barrington-Leigh et al.,
2017). On the other hand, these statistical data are collected indirectly by relevant
statistical departments or road authorities through surveys and data coordination from
various sources (Turner, 2015; CIA, 2025), making it still impossible to accurately
identify whether each road within a country or region is paved or unpaved.
In recent years, with the development of sensing devices, remote sensing, and big
data technologies, many scholars have proposed methods to identify road surface types
based on multiple data sources (Louhghalam et al., 2015; Sattar et al., 2018; Pérez-
Fortes et al., 2022). For example, some scholars have suggested methods using vehicle-
mounted sensing devices to identify road surface types. Chen et al. (2016) designed a
road surface type identification system that can be connected to distributed vehicles and
was tested on 100 taxis in Shenzhen to assess the roughness of road surfaces in
Shenzhen. Harikrishnan et al. (2017) collected vehicle speed data using the XYZ three-
axis accelerometer of smartphones and established road surface type identification



models for four different vehicle speeds. Li and Goldberg (2018) developed a similar
system using smartphones, collecting data from five different drivers over 15 days to
classify road roughness into three categories: "good," "moderate," and "poor". Other
scholars have proposed methods using street view data to identify road surface types.
Randhawa et al. (2025) used a deep learning model combining SWIN-Transformer and
CLIP-based segmentation on Mapillary street-view images to classify road surfaces of
global range into paved and unpaved. Menegazzo et al. (2020) collected street view
data for some roads in Anita Garibaldi, Brazil, using vehicle-mounted cameras and
identified paved and unpaved roads based on a CNN neural network model. Zhou et al.
(2025a) recently utilized crowdsourced street view data from Mapillary to develop a
dataset of road surface type annotations (paved and unpaved) for the African region.
Additionally, some scholars have proposed methods using high-resolution remote
sensing imagery to identify road surface types. Workman et al. (2023) developed a
framework using high-resolution optical satellite imagery and machine learning to
predict the condition of unpaved roads in Tanzania. Zhou et al. (2024) proposed a
method that integrates OpenStreetMap (OSM) and high-resolution Google satellite
imagery to identify road surface types and used this method to develop the road surface
type dataset for Kenya. However, methods based on vehicle-mounted sensing devices
require on-site data collection for each road, inevitably requiring significant manpower,
material, and financial resources, making them difficult to apply to large-scale study
areas such as continents or countries. Data like Google street view are only available in
a few countries or specific regions of countries, making it challenging to identify the



surface types of all roads in a country. Therefore, although the data developed based on
street views covers a global range, it only has 36% of the complete global roads, this
proportion is even lower in Africa and Asia (Randhawa et al., 2025). Remote sensing
methods may suffer from low accuracy in identifying road surface types due to dense
vegetation or building shadows obscuring roads (Zhou et al., 2024). Therefore, Zhou et
al. (2025b) recently proposed a new method based on multisource big data and deep
learning models to infer road surface types, which has been validated in two African
countries. Compared to remote sensing methods, this approach can address the low
accuracy of road surface type identification in areas with poor remote sensing image
quality; for example, the accuracy of remote sensing methods in Cameroon is only 67%,
while the accuracy of the multisource data method in the same region exceeds 85%.
Nevertheless, existing research still has limitations. (1) The method proposed by
Zhou et al. (2025b) has only been validated in a few (1-2) African countries, and it
remains to be verified whether these methods can be applied to develop road surface
type dataset for different African countries. (2) Existing road surface type data are still
mainly statistical data at the national scale, with Zhou et al. (2025b) only providing a
road surface type dataset for Nigeria, leaving a gap in data products covering different
countries and regions in Africa.
Therefore, this study not only aims to evaluate whether the method of developing
road surface type dataset based on multisource big data and deep learning models has
universal applicability but also uses this method to develop the first dataset of road
surface types (paved and unpaved) for 50 countries and regions in Africa. The dataset



developed in this study not only provides information on the surface type of each road
in various countries or regions of Africa but also verifies the accuracy of the dataset:
accuracy ranges from 77% to 96%, and the F1 score ranges from 0.76 to 0.96.
Compared to IRF and other road statistical data, the dataset developed in this study can
support detailed mapping of road surface types in various African countries or regions
and provide data support for road infrastructure construction.
The remainder of this paper is organized as follows: Section 2 introduces the study
area and the source data for developing and evaluating the road surface type data.
Section 3 introduces the methods for data development and evaluation. Section 4
reports the evaluation results of the road surface type data. Section 5 discusses the
implications and limitations of this study. The last two sections provide the data
acquisition methods and the research conclusions.

**2. Study Area and Data**
**2.1 Study area**
This study takes 50 countries and regions in Africa, the second-largest continent on
Earth, as the study area (Figure 1), with a total road length of approximately 6,822,516
kilometers. The main reason for selecting Africa as the study area is that existing
research shows that the proportion of unpaved roads in Africa is high (Biber-
Freudenberger et al., 2025), while the IRF only provides statistics on the length of paved
roads and the road paved rate for some African countries. Due to the lack of spatialized
road surface type dataset, it is difficult to provide decision support for improving road
infrastructure in Africa.

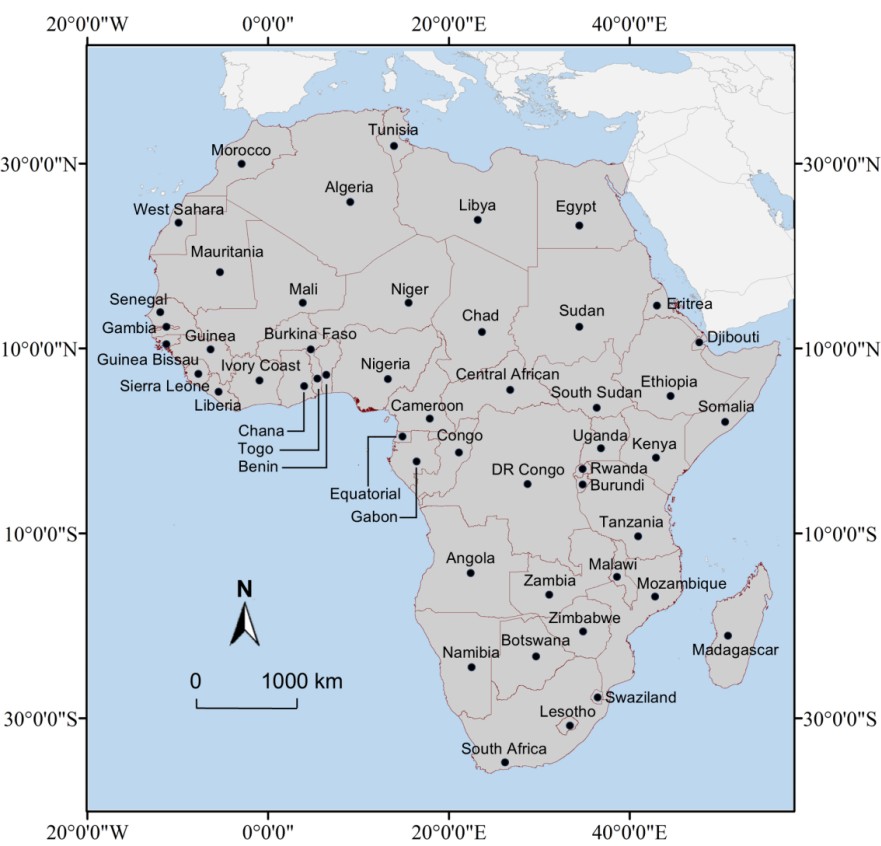


Figure 1. Study area

## 2.2 Data

### 2.2.1 Geospatial data

(1) OpenStreetMap road data: OpenStreetMap (OSM) is an open geospatial dataset
provided online by global volunteers (Harikrishnan et al., 2017). This dataset includes
various geographic elements such as roads, buildings, and water bodies. Each
geographic element not only contains geometric information but also describes its



characteristics or attribute information through a series of tags. Specifically, the
"surface" tag in OSM road data is designed to describe the road surface type of each
road segment. The value of this tag typically refers to the surface material of the road,
such as asphalt, concrete, or gravel. Although OSM data for different countries or
regions in Africa all include road surface type information, incomplete statistics show
that the length of OSM roads with surface type information in a single country usually
accounts for less than 30%, meaning that most OSM road data lack surface type
information, thus urgently requiring supplementation and improvement. This study
obtained road data for 50 countries and regions in Africa (in ESRI Shapefile format)
from the Geofabrik platform (http://download.geofabrik.de/index.html ), which allows
obtaining OSM road data by country.
(2) GDP grid data: This dataset is a 1km spatial resolution GDP grid dataset developed
by Southwestern University of Finance and Economics (Chen et al., 2022). The dataset
was developed by integrating nighttime light remote sensing data (NPP-VIIRS), land
use data, and regional economic statistics using spatial interpolation and machine
learning algorithms. This dataset overcomes the limitations of traditional administrative
unit statistics and can precisely depict the spatial heterogeneity of economic activities.
The dataset spans from 1992 to 2019, and this study used the data from the most recent
year (2019).
(3) Population grid data: This dataset is the LandScan global population dataset
developed by Oak Ridge National Laboratory (ORNL) in the United States, with a
spatial resolution of 30 arc seconds in latitude and longitude (approximately 1km at the
equator) (Dobson et al., 2000). The dataset integrates census data, satellite imagery, and
mobile communication data, using dynamic modeling methods to simulate 24-hour
population distribution. Existing research has found that compared to other population
grid datasets (such as WorldPop and Global Human Settlement Population Grid),
LandScan has higher accuracy (Jiang et al., 2021; Mohit et al., 2021; Yin et al., 2021).
Therefore, this study obtained the 2020 LandScan population raster data for the African
region (https://landscan.ornl.gov/).
(4) Building height data: This dataset is a 100-meter resolution building height dataset
released by the Global Human Settlement Layer (GHSL). The dataset is based on
Sentinel-1/2 and Landsat imagery, using machine learning algorithms to extract the
three-dimensional morphology of buildings (Pesaresi et al., 2021). The dataset includes
building height raster data. GHSL-BUILT is the world's first building height dataset,
and this study obtained the 2018 building height data recommended by GHSL for
analysis (https://human-settlement.emergency.copernicus.eu/ghs_buH2023.php).
(5) Land cover data: This dataset is a global land cover dataset with a 10-meter spatial
resolution released by ESRI. The dataset was developed based on Sentinel-2 imagery
and deep learning methods, including nine different land cover categories (water, trees,
flooded vegetation, crops, buildings, bare land, snow, clouds, and pasture) (Karra et al.,
2021). Existing research indicates that ESRI land cover data has better accuracy
compared to other similar datasets (such as ESA World Cover and Dynamic World)
(Yan et al., 2023). This study obtained the 2020 Land Cover data for the African region
(https://livingatlas.arcgis.com/landcover/).



**2.2.2 Statistical data**
To verify the effectiveness of the data, this study also obtained two types of
statistical data, IRF road statistics and socio-economic statistics.
(1) IRF Road Statistics: The International Road Federation (IRF) is a non-profit
international organization dedicated to promoting development and cooperation in the
global road transport sector (Turner, 2015). IRF provides free and rich statistical data
resources to global users (https://www.irf.global/). These data primarily come from
authoritative reports and statistical agencies of various governments, covering multiple
fields such as road networks and the transportation industry. This study obtained three
statistical data provided by IRF for the African region in 2020, namely the length of
paved roads, total road length, and road paved rate.
(2) Socioeconomic Statistics: Existing research has found that the road paved rate is
highly positively correlated with the level of socioeconomic development (Anyanwu et
al., 2009). Therefore, this study also introduced two indicators related to the level of
socioeconomic development, namely the Human Development Index (HDI) and Gross
National Income per capita (GNI per capita, based on PPP current international $). HDI
is compiled and published by the United Nations Development Programme since 1990,
obtained by comprehensively evaluating a country's life expectancy, average years of
schooling, and gross national income, and is used to measure the socioeconomic
development level of various countries. GNI per capita is published by the World Bank,
where GNI is the sum of the incomes of all residents in a country or region; GNI per
capita is the average GNI of a country or region, which can measure the average



economic income level of the nationals in a country or region. This study obtained the
2020 HDI and GNI per capita data, covering 44 and 36 African countries and regions,
respectively.

**3. Methods**
The technical roadmap of this study is shown in Figure 2.

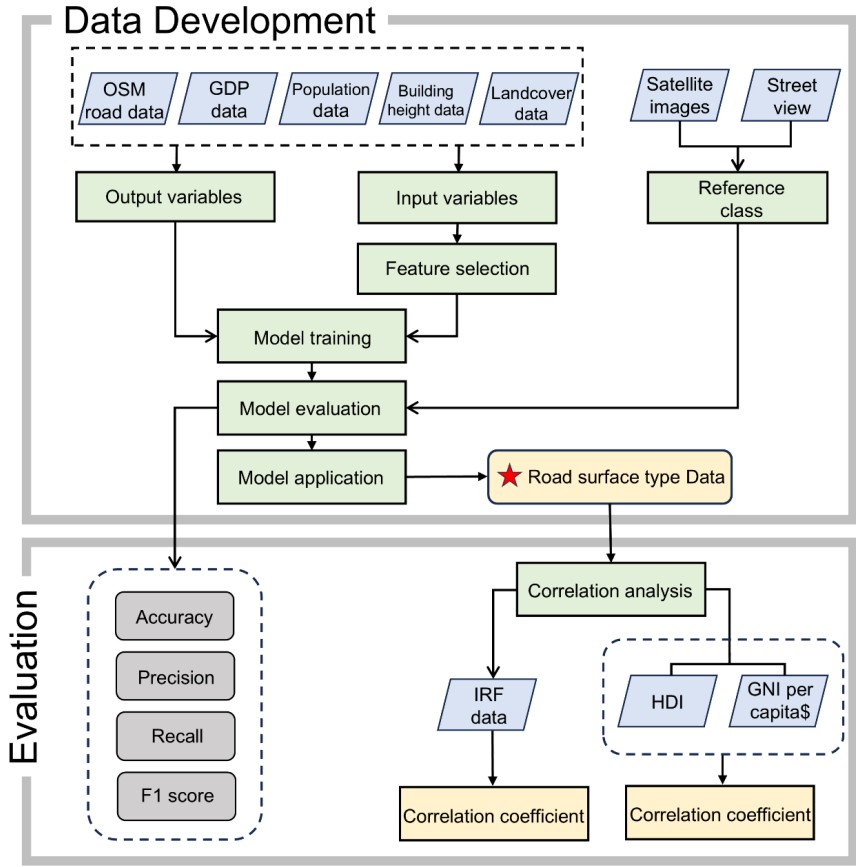


Figure 2. Technical roadmap

**3.1 Developing of Road Surface Type Dataset of Africa**



This study utilizes a method recently proposed by Zhou et al. (2025b) that is based
on multi-source geospatial big data and deep learning models to develop the road
surface type dataset of 50 African countries and regions. The main idea of this method
includes the following steps: First, sampling points and corresponding OpenStreetMap
(OSM) road surface type labels are acquired based on OSM road data. Then, proxy
indicators that characterize road surface types are calculated based on multi-source
open geospatial big data. Third, a deep learning model is trained using the proxy
indicators and road surface type labels of the sampling points. Finally, the trained model
is applied to the road networks of various African countries and regions to identify the
surface type of each road.
**3.1.1 Road Sampling**
According to the definition of OSM road level tags (highway=) as outlined in the
OSM wiki (https://wiki.openstreetmap.org/wiki/Key:highway), roads passable by four-
wheeled motor vehicles are selected. These specifically include: "highway= motorway,
motorway_link, trunk, trunk_link, primary, primary_link, secondary, secondary_link,
tertiary, tertiary_link, residential, living_street, service, track, road, unclassified". Other
roads primarily intended for bicycles or pedestrians (e.g., cycleway, footway) are
excluded from the analysis.
After that, the selected OSM road data are then sampled at 100-meter intervals to
generate sampling points. The 100-meter interval is chosen because most roads are
greater than or equal to 100 meters in length, ensuring that most roads have at least one
sampling point. For roads shorter than 100 meters, the center point of the road is used

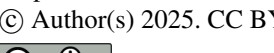



as the sampling point.
**3.1.2 Calculation and Selection of Proxy Indicators**
(1) Calculation of Proxy Indicators
It has been found by Zhou et al. (2025b) that road surface types are not only related
to road classes but also to the socio-economic and geographical environment of the area
where the road is located. Therefore, Zhou et al. (2025b) designed 16 proxy indicators
across three feature dimensions—Road network features, Socio-economic features, and
Geographical environment features—as shown in Table 1. These indicators serve as
"proxies" to identify or infer road surface types.



Table 1. Proxy Indicators

| Dimension | Data Source | No. | Input | Type |
|---|---|---|---|---|
| Road network features | OSM road data | 1 | Road class | Category |
| | | 2 | Road length | Value |
| | | 3 | Degree | |
| | | 4 | Closeness | |
| | | 5 | Betweenness | |
| Socio-economic features | GDP | 6 | GDP | Value |
| | Population | 7 | Population | |
| | Building height | 8 | Building height | |
| Geographical environment features | Land cover | 9 | Water proportion | Value |
| | | 10 | Trees proportion | |
| | | 11 | Flooded vegetation proportion | |
| | | 12 | Crops proportion | |
| | | 13 | Building proportion | |
| | | 14 | Bare land proportion | |
| | | 15 | Snow land proportion | |
| | | 16 | Pasture proportion | |


For a single road sampling point,
Road network features: The road class is directly obtained from the OSM
"highway=" tag. To calculate road length, degree centrality (Degree), closeness
centrality (Closeness), and betweenness centrality (Betweenness). The road networks
of each country or region are constructed into strokes based on the "every best fit" rule.
These metrics (road length, Degree, Closeness, Betweenness) are calculated for each



stroke. The values are assigned to the corresponding sampling points on the road (Zhou
et al., 2012).

Socio-economic features: The sampling point is assigned the value of the grid cell

it falls into for corresponding data (GDP, population, or building height).

Geographical environment features: A 100m x 100m grid unit is established. The

sampling point's grid unit is identified. The proportion of each land cover type within
that grid unit is calculated.
(2) Feature Selection

Since proxy indicators may be highly correlated, this study employs correlation

analysis and contribution analysis to select appropriate proxy indicators for model
training, aiming to reduce data dimensionality, simplify model complexity, and
eliminate multicollinearity.

For a single country or region: First, the correlation between pairs of proxy

indicators is calculated using Phi_k (Baak et al., 2020), chosen because it can measure
the correlation coefficient between different types of variables. Second, Shapley
Additive exPlanations (SHAP) are used to analyze the interpretability of each proxy
indicator, quantifying its contribution to the model's predictions. Third, proxy
indicators without multicollinearity are directly used as input features. If two proxy
indicators exhibit multicollinearity, the one with the highest contribution (based on
SHAP values) is retained as the input feature for that country or region.
(3) Road surface type classification

Road surface types are treated as output variables and defined into two categories





based on whether the road is paved. Paved roads: roads with a structured surface.
Unpaved roads: roads without a structured surface.
Since the labels for training samples are automatically extracted from the OSM
"surface=" tag, all OSM tags are reclassified into "paved" or "unpaved" roads, as shown
in Table 2. The reclassification criteria follow the guidelines provided by OSM's wiki
(https://wiki.openstreetmap.org/wiki/Surface ).
Table 2. Reclassifying OSM "surface=" tags into paved and unpaved roads.

| OSM "surface=" Tag | Reclassification |
|---|---|
| Asphalt, Concrete, Concrete: Plates, Paved, Paving Stones, Sett | Paved |
| Compacted, Dirt, Earth, Fine_Gravel, Gravel, Ground, Mud, Pebblestone, Sand, Unpaved | Unpaved |


**3.1.3 Model Training and Application**
Zhou et al. (2025b) compared six machine learning and deep learning models for
identifying road surface types and found that the TabNet model achieved the highest
accuracy (approximately 86%). Consequently, this study adopts TabNet to develop the
road surface type dataset for 50 African countries and regions. TabNet, proposed by
Arik et al. (2021), combines the end-to-end learning and representation learning
characteristics of deep neural networks (DNNs) with the interpretability and sparse
feature selection advantages of decision tree models.





329 For a single African country: From sampling points with "surface=" tags, 5000

330 paved and 5000 unpaved sampling points are randomly selected as training samples. In

331 some countries or regions where the number of paved sampling points is less than 5000

332 (e.g., a minimum of approximately 3000), all paved sampling points (e.g., 3000) and

333 an equal number of unpaved sampling points (e.g., 3000) are used.

334 For each training sample, the 16 proxy indicators from Table 1 are calculated. After

335 feature selection, the selected proxy indicators serve as input features for model training.

336 The OSM road surface type of the training sample is used as the model output. The

337 TabNet model is trained, with parameters (e.g., learning rate, batch size, training epoch)

338 automatically determined using the Optuna framework, which searches for optimal

339 parameters during training.

340 Each country trains a separate model. The trained model infers the road surface

341 type of each sampling point in that country. A correction strategy proposed by Zhou et

342 al. (2025b) is applied to determine the final surface type of each road segment, where

343 the surface type is determined by the majority surface type of its sampling points.

345 **3.2 Result evaluation**

346 This study evaluates the effectiveness of the developed road surface type dataset

347 from three aspects.

348 **3.2.1 Accuracy assessment**

349 For each African country or region: From all sampling points (excluding training

350 samples), 500 points predicted as "paved" and 500 predicted as "unpaved" are randomly



selected, totaling 1000 validation points. Three different operators visually interpret the
classification results of each validation point using high-resolution Google satellite
imagery and Google street view, with the final reference surface type determined by
voting.
At last, the model's predictions are compared with the reference road surface types,
and effectiveness is assessed by calculating accuracy, precision, recall, and F1 score.

**3.2.2 Comparative evaluation with existing statistical data**

Based on the developed road surface type dataset, the paved road length, total road
length, and road paved rate for each country and region are calculated and compared
with International Road Federation (IRF) statistical data. Specifically, correlation
coefficients between the results calculated from this data product and IRF statistical
values are explored.
Since IRF provided statistical values for only 19 African countries in 2020, only
these 19 countries are included in the correlation analysis.

**3.2.3 Correlation evaluation with socio-economic indicators**

Existing research indicates that the road paved rate is highly positively correlated
with socio-economic development levels (Anyanwu et al., 2009). Therefore, this study
explores the correlation between the road paved rate calculated from this data product
and two indicators: Human Development Index (HDI), Gross National Income per
capita (GNI per capita, based on PPP current international $).
More precisely, the analysis includes 44 African countries with HDI data and 36
with GNI per capita statistical data to verify the effectiveness of the data product.

## 4. Results and Analyses

### 4.1 Description of the Africa Road Surface Type Dataset

This study has developed the road surface type dataset that records the roads and

its surface type attribute information for 50 African countries and regions, as shown in

Figure 3.

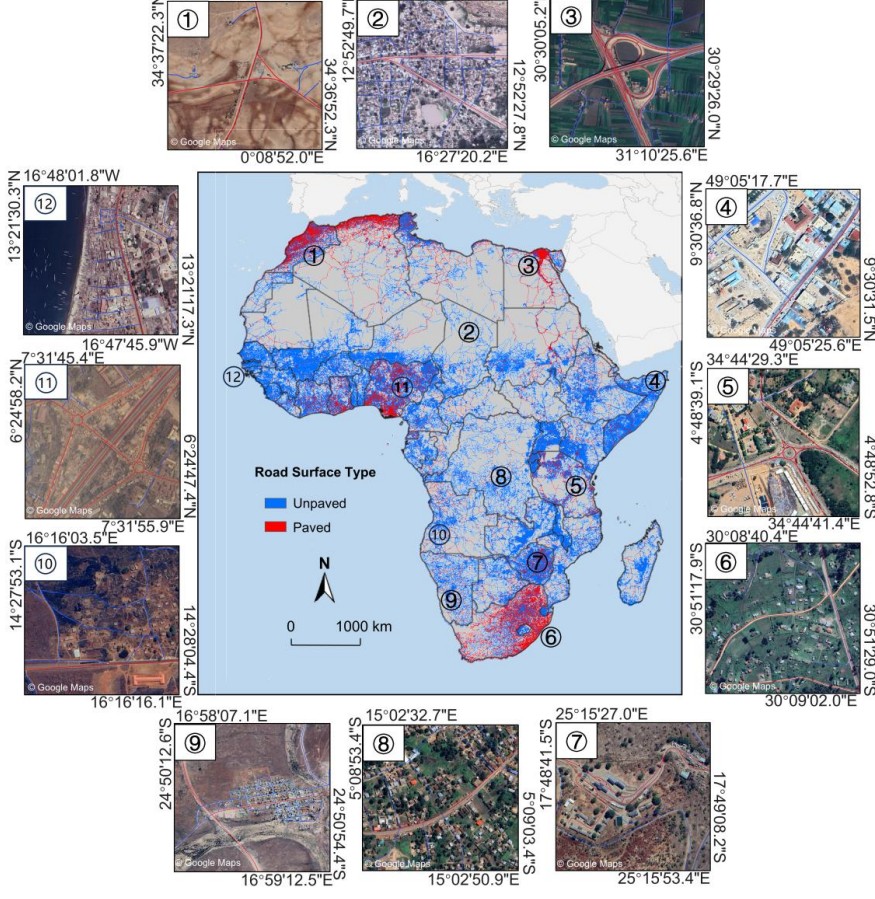

Figure 3. Visualization of road surface type dataset for 50 African countries and

regions (source: Google Maps. 2025, https://www.google.com/maps/ (last access: 2




382         Jul 2025))

383  This dataset was developed based on OpenStreetMap (OSM) road data for Africa,

with each country and region stored as a separate vector file in ESRI Shapefile format,
using the WGS 1984 Web Mercator projection. The road data for each country and
region includes five attribute fields: road ID, coordinates of the start and end points
(Table 3), road length, and road surface type. The entire dataset comprises
approximately 13,309,000 road segments, with a total length of about 6,822,516 km.

389        Table 3. Descriptions of dataset

| Attribute | Description | Type |
|---|---|---|
| ID | Road segment ID | Int |
| Start point | Coordinates of the road segment's start point (x, y) | String |
| End point | Coordinates of the road segment's end point (x, y) | String |
| Road length | Length of the road segment (calculated based on WGS 1984 Web Mercator) | Float |
| Surface type | Road surface type, i.e., paved or unpaved | String |


**4.2 Accuracy Assessment of the Road Surface Type Identification Model**

392  The accuracy assessment results of the road surface type dataset for 50 African

countries and regions are presented in Figure 4. As indicated in the figure, the average
accuracy across the 50 countries and regions is 86.8%. Out of these, 44 countries and
regions have an accuracy above 80%, and 12 out of 50 have an accuracy exceeding
90%. The country with the highest accuracy is Burundi, surpassing 96%, while the
lowest is Egypt, at approximately 77%.

For paved roads, the average precision, recall, and F1 score across the 50 countries

and regions are 88.0%, 85.0%, and 0.86, respectively. Specifically, 45 countries and
regions have a precision above 80%, 32 have a recall above 80%, and 43 have an F1
score above 0.80 for paved roads.

For unpaved roads, the average precision, recall, and F1 score are 86.3%, 88.2%,

and 0.87, respectively. Among the 50 countries and regions, 36 have a precision above
80%, 46 have a recall above 80%, and 46 have an F1 score above 0.80 for unpaved
roads.

These results demonstrate that the road surface type dataset developed in this study

has relatively high accuracy, consistent with the accuracy reported in existing research
(approximately 86%) (Zhou et al., 2025b), indicating that the method using multi-
source geospatial big data and deep learning models for identifying road surface types
has certain universality.

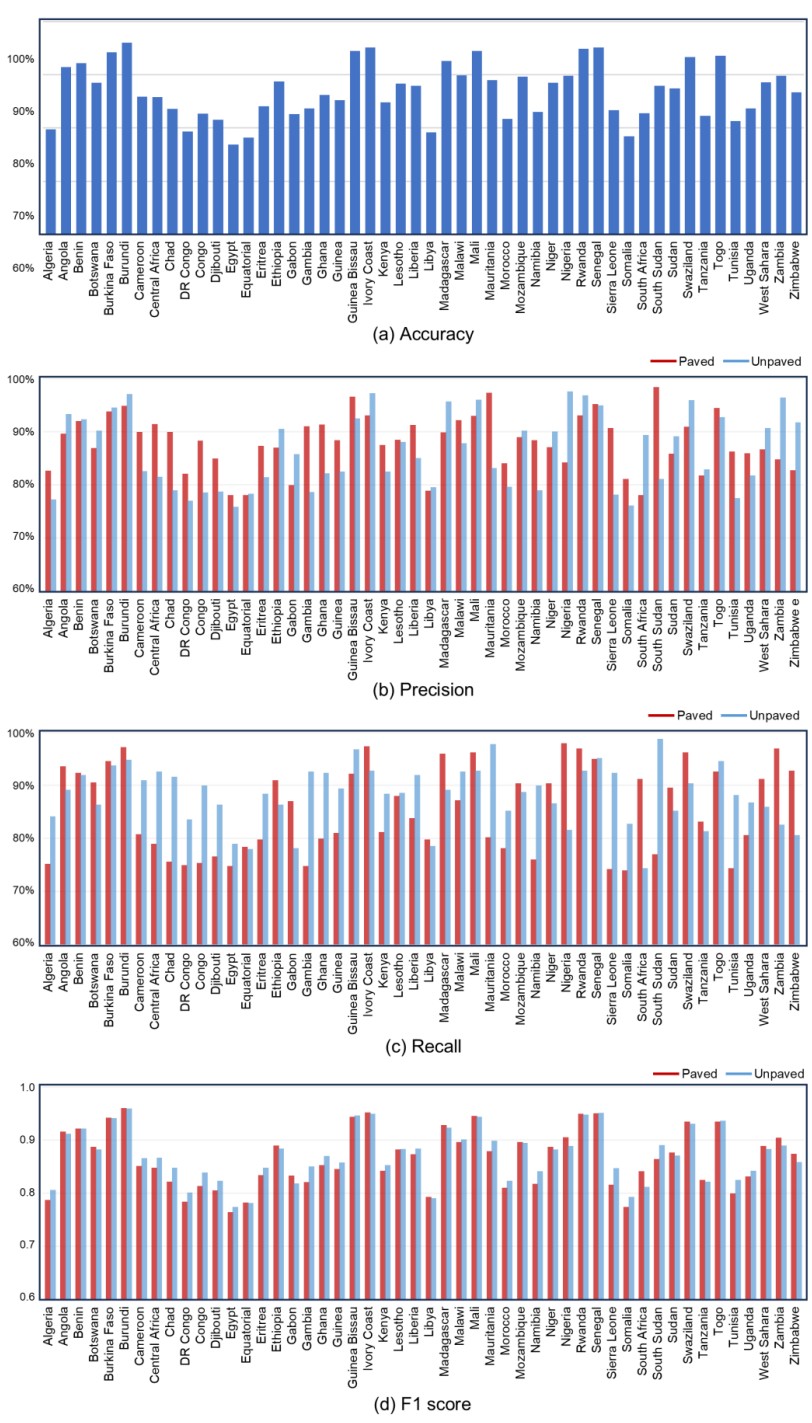


Figure 4. Accuracy Assessment Results of the Road Surface Type Dataset




### 4.3 Comparative Assessment with IRF Statistical Data


Figure 5 presents the correlation analysis results between the total road length,
paved road length, and road paved rate calculated based on the road surface type dataset
developed in this study and the corresponding statistical data from the International
Road Federation (IRF).
The correlation coefficients for total road length, paved road length, and road paved
rate are 0.89, 0.94, and 0.69, respectively, all indicating a high correlation. This suggests
that the calculations based on our data product are generally consistent with the IRF
statistical data in terms of trends. For example, South Africa has the longest total road
length and paved road length, while Gambia has the shortest; Tunisia and Morocco have
the highest road paved rates. These results indicate the rationality of the road surface
type dataset.
However, as shown in the scatter plots (Figure 5), there are still discrepancies
between the calculations based on our data product and the IRF statistical data.
Specifically, the total road length calculated from our data product is consistently higher
than that reported by IRF (as seen in Figure 5a, where points are located to the left of
the diagonal). Similarly, for 18 out of 19 countries, the paved road length is higher than
the IRF statistics. Existing research has pointed out that IRF statistical data may
underestimate the total road length globally, with an average underestimation of 36%,
and for 94 countries, the underestimation exceeds 50% (Barrington-Leigh et al., 2017).
Therefore, IRF statistical data may underestimate the total road length and paved road
length in African countries.
Additionally, for 15 out of 19 countries, the road paved rate is lower than that
reported by IRF. This may be because IRF data underestimates the total road length in
African countries, and the unaccounted roads are likely mostly unpaved, leading to an
overestimation of the road paved rate in IRF statistics.

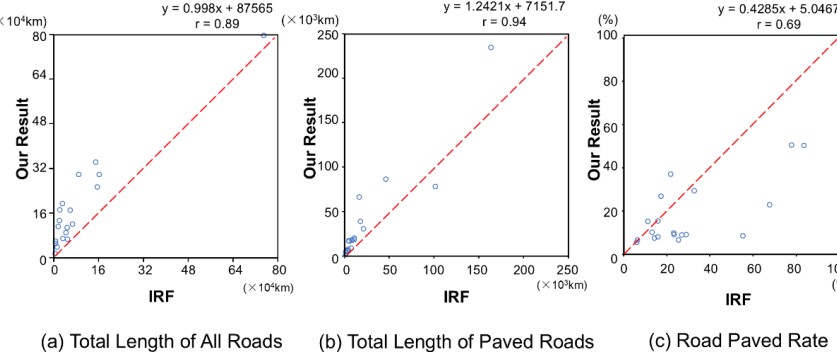

(a) Total Length of All Roads        (b) Total Length of Paved Roads        (c) Road Paved Rate


Figure 5. The Correlation Analysis Results with IRF Statistical Data

**4.4 Correlation Assessment with Socioeconomic Indicators**
The correlation analysis results between the road paved rate calculated based on
our data product for 50 African countries and regions and the Gross National Income
per capita (GNI per capita) and the Human Development Index (HDI) are shown in
Figure 6. As indicated, the correlation coefficients between the road paved rate and GNI
per capita and HDI are 0.80 and 0.83, respectively, both showing a strong positive
correlation. This indicates that the road paved rate in African countries is highly
positively correlated with their level of socioeconomic development, consistent with
findings from existing research (Anyanwu et al., 2009), indirectly validating the
effectiveness of our road surface type dataset.

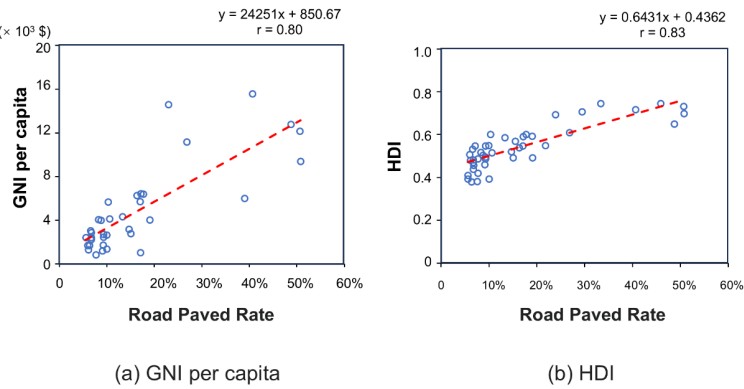

(a) GNI per capita                    (b) HDI


Figure 6. The Correlation Analysis Results of The Road Paved Rate Calculated Based
on The African Road surface type dataset with Per Capita GNI (a) and HDI (b)

**4.5 Spatial Pattern Analysis of Road Paved Rates in Africa**

Based on the road surface type dataset, the spatial patterns of road paved rates in

50 African countries and regions were analyzed at the national, provincial, and county
levels, as shown in Figure 7. Compared to IRF, which only provides statistical data for
19 African countries (Ken et al., 2008), our dataset not only allows for the analysis of
road paved rates in all 50 African countries and regions but also enables detailed
analysis at different administrative levels.

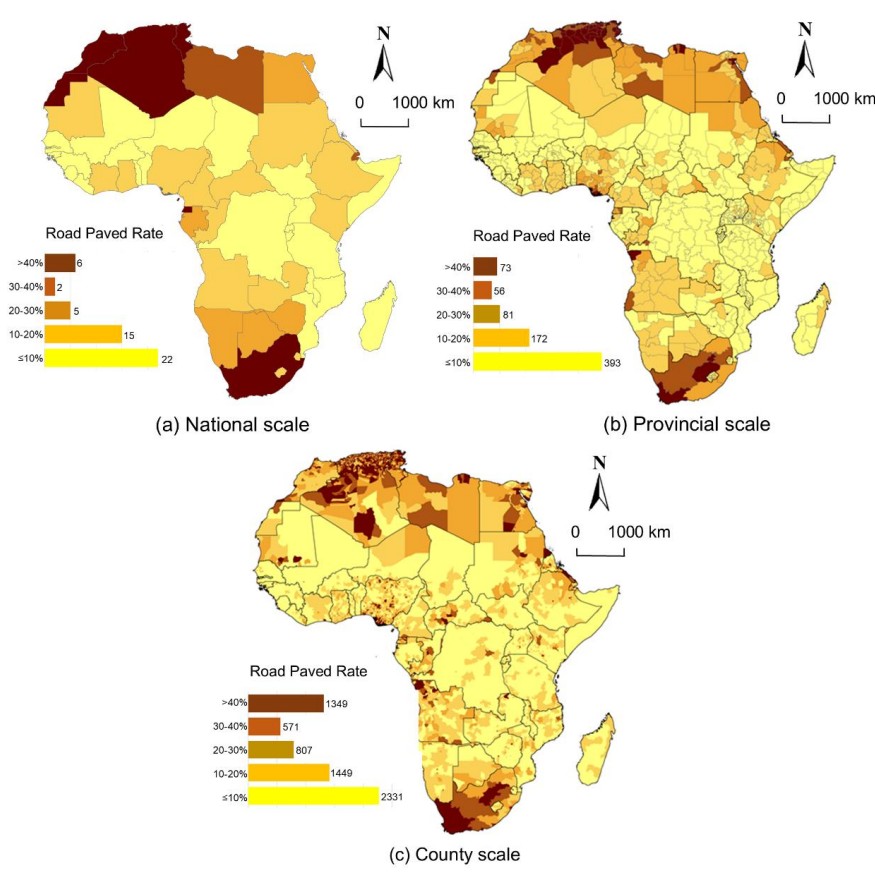


Figure 7. Spatial Pattern Analysis at National, Provincial, and County Levels


467  At the national level, the average road paved rate across the 50 African countries

468 and regions is only 17.4%, ranging from a low of 5.54% in Chad to a high of 50.77%

469 in Morocco. Only six African countries have a road paved rate above 40%, while 37

470 countries and regions have a rate below 20%. The average road paved rate for 43

471 countries and regions in Sub-Saharan (excluding South Africa) is merely 13.6%. These

472 results indicate that road paved rates in African countries and regions are generally low,

473 with significant north-south disparities. At the provincial and county levels, only 9% of





provincial administrative divisions have a road paved rate above 40%, mostly located
in north of Africa and South Africa. Similarly, only about 20% of county administrative
divisions have a road paved rate above 40%, primarily in north of Africa, South Africa,
and some urban areas. Therefore, the overall spatial pattern of road paved rates in Africa
shows a " higher in the north and south, lower in the central region " distribution, with
higher rates in north of Africa and South Africa, and lower rates in Sub-Saharan
excluding South Africa. The average road paved rate in the north of Africa (40.7%) is
approximately three times that of Sub-Saharan (excluding South Africa).

**5. Discussion**
**5.1 Data Quality**

This study developed road surface type dataset for 50 African countries and regions

and verified its validity (accuracy ranging from 77% to 96%; F1 score ranging from
0.76 to 0.96). However, the quality of the dataset varies across different African
countries and regions. For example, Burundi has an accuracy of 96%, while Egypt's
accuracy is only 77%. Further, taking a local area in Egypt as an example, combined
with Google high-resolution remote sensing imagery and Google street view, it can be
observed that the backbone of the road network in this region predominantly consists
of paved roads (Figure 8b), while non-backbone roads (especially in rural areas) are
mostly unpaved (Figure 8c); urban areas in Egypt are predominantly paved (Figure 8d),
although some roads remain unpaved (Figure 8e). These results indicate that the road
surface type classification in this study is reasonable.

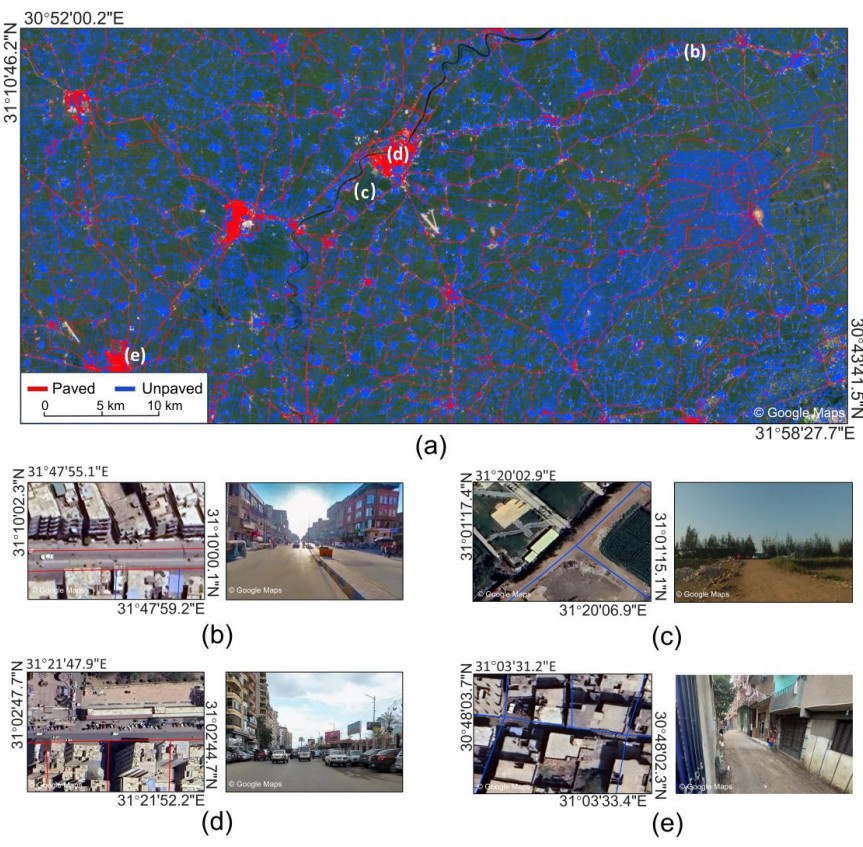

Figure 8. An Example of Road Surface Type Data in Egypt (source: Google Maps.

2025, https://www.google.com/maps/ (last access: 2 Jul 2025))

Despite this, we found that misclassifications of road surface types are inevitable. Taking urban areas in Egypt as an example (Figure 9a), Figure 9b shows a 1 km × 1 km grid area in this region. Figure 9c displays two road classes in this grid area: "trunk" and "residential." From Figures 9b and 9c, it can be seen that most "trunk" roads in this area are classified as paved, while most "residential" roads are classified as unpaved. However, based on street view imagery of this area, it is evident that "residential" roads include both unpaved (Figure 9d) and paved (Figure 9e) types. Therefore, it is difficult

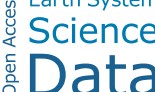

507 to distinguish road surface types in this area based solely on road class, and the spatial

508 resolution of the GDP and population data we obtained (both 1 km) also makes it

509 challenging to finely differentiate road surface types within this area.

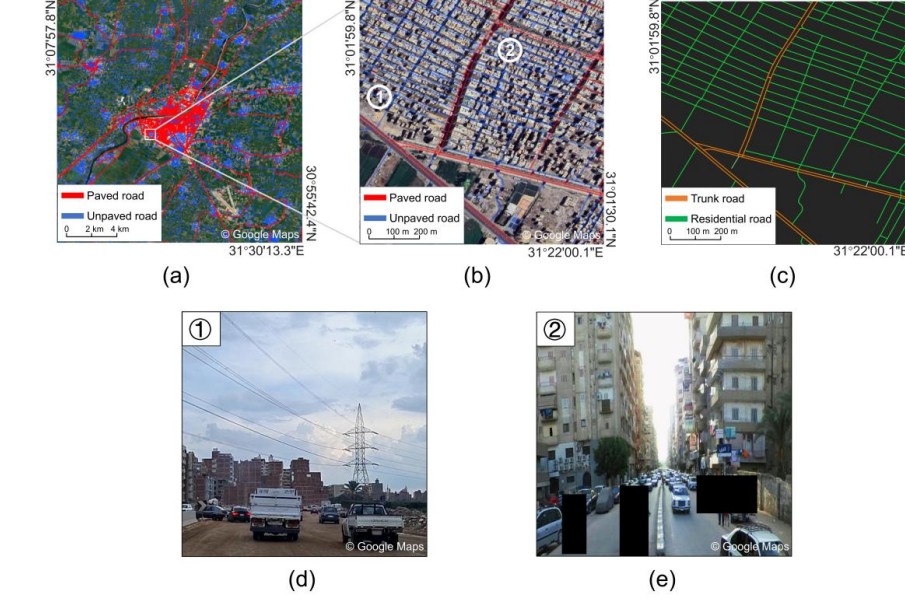

511 Figure 9. An Example of Explaining the Data Quality of The African Road surface

512 type dataset (source: Google Maps. 2025, https://www.google.com/maps/ (last access:

513 2 Jul 2025) )

514 Additionally, open geospatial data inevitably have quality issues. For instance,

515 although existing studies have found that the geometric positional accuracy and

516 completeness of OSM road data in Africa are generally high, road data gaps are

517 unavoidable (Zhou et al., 2022); road surface types and road classes labeled by global

518 volunteers in OSM may also contain errors (Zhou et al., 2022). The GHSL-BUILT

519 building height data, derived from medium-resolution remote sensing imagery

520 (Sentinel-2), also inevitably has estimation biases for building heights (Pesaresi et al.,





2021)[34]. LandScan data may be underestimated in urban-rural transition zones and
overestimated in sparsely populated areas (Beata et al., 2019). Nevertheless, OSM road
data remain the only globally available open data source that includes road surface type
labels; GHSL and LandScan data are also globally covered, freely accessible geospatial
data products with long time series, which is why this study selected these data for
experimental analysis. However, in the future, other data sources (e.g., CORINE Land
Cover (Pontius Jr et al., 2017), World Settlement Footprint (Marconcini et al., 2020),
and Global Human Settlement Population Grid (Yin et al., 2021)) could be considered,
and their impact on the quality of road surface type dataset could be analyzed.

**5.2 Implications and Significance**
Compared to traditional statistical data such as those from IRF, the first-ever road
surface type dataset for 50 African countries and regions developed in this study not
only allows for the calculation of statistical indicators such as paved road length and
road paved rate for each country and region but also enables detailed analysis of which
roads are paved or unpaved, providing decision-making support for improving local
transportation infrastructure (e.g., upgrading unpaved roads to paved roads).
Additionally, road surface types are an important data source for assessing SDG 9.1.
Therefore, this dataset can also be combined with population and urban built-up area
data to analyze the proportion of rural populations within 2 km of paved or unpaved
roads in various African countries (Wanjing et al., 2021), to provide data support for
evaluating Africa's sustainable development goals. Last but not least, this dataset can

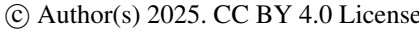

be combined with location data of traffic accidents to analyze the relationship between
road surface types and traffic accidents (Patrick et al., 2022); with traffic carbon
emission data to analyze the relationship between road surface types and environmental
impacts (Ling et al., 2024); or with national income data to analyze the relationship
between road surface types and socioeconomic development (Anyanwu et al., 2009).

Moreover, this study utilized multisource geospatial big data and deep learning

models to develop the African road surface type dataset. The primary advantage of this
method is that its source data (including OSM, LandScan, GDP, GHSL-BUILT, and
ESRI Land Cover) are not only openly accessible but also globally covered. Therefore,
this method could also be applied to identify road surface types in other countries and
regions worldwide, providing methodological support for developing global road
surface type dataset.
**5.3 Limitations and future work**

(1) This study adopted the method proposed by Zhou et al. (2025b) to develop the

African road surface type dataset. This method designs 16 proxy indicators across three
dimensions (Road network, Socioeconomic, and Geographical Environment) from five
types of open geospatial data to infer road surface types. In the future, other data sources
such as terrain data could be introduced, and additional proxy indicators such as slope,
aspect, and surface roughness could be designed to investigate whether these indicators
can improve the classification accuracy of the data product.

(2) Road surface types are not limited to just paved and unpaved roads; they can

also be further subdivided into categories such as asphalt, concrete, and dirt roads.



However, we found that most paved roads in Africa are asphalt roads, and most unpaved
roads are dirt roads; thus, this study only considered "paved" and "unpaved" categories.
Nevertheless, in the future, by supplementing field-measured data, it could be explored
whether this method can be used to develop dataset that include more detailed road
surface type classifications.
(3) The African road surface type dataset developed in this study is limited to a
single year, approximately 2020. This is because the source data used were all obtained
from 2020 or nearby years to ensure temporal consistency across dataset for different
African countries. Although most open geospatial big data (such as OSM, GDP, and
population data) include data from different years, which could potentially be used to
develop road surface type dataset for multiple years, validation data are difficult to
obtain. Specifically, it is challenging to interpret roads and their surface types using
open-source medium- to low-resolution satellite imagery (e.g., Landsat or Sentinel-2).
Although Google satellite imagery has higher resolution, the update years of Google
imagery for different areas within a country may not be consistent, making it difficult
to analyze changes in road surface types. Nonetheless, in the future, this method could
be attempted to develop road surface type dataset for different years, and accuracy could
be validated using long-time-series high-resolution remote sensing imagery; further,
spatiotemporal changes in road surface types at a large scale could be analyzed.

**6. Data availability**
The First Road Surface Dataset for 50 African countries and reigns is distributed



under the CC BY 4.0 License. The data can be downloaded from the data repository
Figshare at https://doi.org/10.6084/m9.figshare.29424107 (Liu et al., 2025).
**7. Conclusion**
This study developed the first dataset containing road surface types for every road
in 50 African countries and regions, based on multi-source geospatial data and deep
learning model. The accuracy of this dataset was evaluated through visual interpretation
using high-resolution Google satellite imagery and Google street view, while its
effectiveness was indirectly analyzed by comparing it with IRF statistical data and
socio-economic indicators such as HDI and GNI per capita. Finally, the spatial patterns
of road surface types across these 50 African countries and regions were analyzed using
the developed dataset. The main findings are as follows:
(1) The accuracy of the road surface type dataset for the 50 African countries and
regions ranges from 77% to 96%, with F1 scores between 0.76 and 0.96, validating the
effectiveness of the developed dataset.
(2) In terms of total road length, paved road length, and road paved rate, the
correlation coefficients between the calculations based on our dataset and the IRF
statistical data show high correlation, ranging from 0.69 to 0.94. Regarding socio-
economic indicators (GNI per capita and HDI), the calculations based on our dataset
also exhibit high correlation with the relevant statistical data, ranging from 0.80 to 0.83,
indirectly verifying the effectiveness of our dataset.
(3) From a spatial perspective, the road paved rate in Africa is generally low. The
average road paved rate across the 50 African countries and regions is only 17.4%,





displaying a spatial pattern of "higher in the north and south, lower in the central
region." Specifically, the average road paved rate in the north of Saharan is
approximately 3 times that of Sub-Saharan (excluding South Africa).
The dataset developed in this study includes the surface type of every road in Africa,
offering decision-making support for improving the region's road infrastructure.
Additionally, this dataset can be combined with data on population and urban built-up
areas to assess Africa's Sustainable Development Goals (e.g., SDG 9.1). Furthermore,
it can be integrated with other datasets—such as traffic accidents, carbon emissions,
and national income—to analyze the impact of road surface types on road safety, energy
consumption, ecological environment, and socio-economic development.

**Author contributions** ZL developed the data and wrote the original manuscript. QZ
proposed methods and designed experiments. FZ reviewed and improved the
manuscript. LP checked and validated data quality. All authors discussed and improved
the manuscript.

**Competing interests** The contact author has declared that none of the authors has
any competing interests.
**Acknowledgements** The project was supported by National Natural Science
Foundation of China (Grant No. 42471492).





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
