# Peer review of "The First Road Surface Type Dataset for 50 African"

_Earth System Science Data, 2025_

## Author Comment (AC1)

**Response to Reviewer 1**

**Reviewer1_Comment1**

In Figure 1, it appears that both the input and output variables point to model training, which could be somewhat confusing. Could the authors provide further explanation and clarification??

**Reply to Reviewer1_Comment1:** Thanks for this valuable comment! We have revised "Output variables" as "Road surface types"; and revised "Input variables" as "Proxy indicators" (see **Figure 2**). We have explained both the "Road surface types" and "Proxy indicators" in the revised manuscript. That is,

"Therefore, Zhou et al. (2025b) designed 16 proxy indicators across three feature dimensions—Road network features, Socio-economic features, and Geographical environment features—as shown in Table 1. These indicators serve as "proxies" to identify or infer road surface types." (see **Section 3.1.2**)

"Road surface types from OSM data are treated as output variables and defined into two categories based on whether the road is paved. Paved roads: roads with a structured surface. Unpaved roads: roads without a structured surface." (see **Section 3.1.2**)

[Figure]

Figure 2. Technical roadmap

**Reviewer1_Comment2**

In Lines 94-97, there are two ″in Shenzhen″.

**Reply to Reviewer1_Comment2:** Thanks for this valuable comment!

We have removed one of the "in Shenzhen". The sentence was revised as: "Chen et

al. (2016) designed a road surface type identification system that can be connected to

distributed vehicles and was tested on 100 taxis **in Shenzhen** to assess the roughness of road surfaces." (see **Section Introduction**)

**Reviewer1_Comment3**

In Line 285, would it be more appropriate to replace the comma in "For a single road sampling point," with a colon, i.e., "For a single road sampling point:"?

**Reply to Reviewer1_Comment3:** Thanks for this valuable comment!

In the manuscript, we have revised "For a single road sampling point," as "For a single road sampling point:".

**Reviewer1_Comment4**

In Line 289, to improve clarity and readability, please explain the "every best fit" rule and how it is implemented.

**Reply to Reviewer1_Comment4:** Thanks for this valuable comment!

The "every best fit" method, which means to concatenate road segments into individual roads (called strokes). The method has been widely used for street network analysis (Biljecki et al., 2021; Noori et al., 2020) and map generalization (Zhou et al., 2012).

The principle of this method is to connect road segments with the smallest deflection angle. For example, there are three road segments (i.e., Segment 1, Segment 2, and Segment 3 in Figure A). Among these road segments, the angle α is smaller than

the angle β, and thus the Segment 1 should be connected to Segment 2. After that, the

three segments were divided into two "strokes" (e.g., Stroke 1 and Stroke 2).

[Figure]

Figure A The principle of "Stroke" building.

In the revised manuscript, we have highlighted the above point. That is,

"To calculate road length, degree centrality (Degree), closeness centrality (Closeness),

and betweenness centrality (Betweenness), the road networks of each country or region

are constructed into strokes based on the "every best fit" method (Zhou et al., 2012).

**The core principle of this method is to connect continuous road segments into**

**individual roads (called "strokes"), according to the deflection angle between**

**adjacent road segments**." (see **Section 3.1.2**)

Reference:

1) Biljecki, F., Ito, K. 2021. An open-source tool to extract natural continuity and hierarchy of urban street networks. Environment and Planning B: Urban Analytics and City Science, 48(2): 218–237.

2) Noori, N., Tamim, I., Godin, C., Poulin, P. 2020. A deep learning approach to urban street functionality prediction based on centrality measures and urban indicators. Computers, Environment and Urban Systems, 84: 101523.

3) Zhou, Q., Li, Z. 2012. A comparative study of various strategies to concatenate road segments into strokes for map generalization. International Journal of Geographical Information Science, 26(4): 691–715.

**Reviewer1_Comment5**

In Line 315, the authors used the "surface=" tags in OSM data for road types. Are the existing road types in OSM incomplete? Does this study help to address or fill this gap?

**Reply to Reviewer1_Comment5:** Thanks for this valuable comment!

(1) Are the existing road types in OSM incomplete?

Yes. To our knowledge, the length of OSM roads with surface type information in a single country usually accounts for **less than 30%**. We have highlighted this point in the revised manuscript. That is,

"Although OSM data for different countries or regions in Africa include information on road surface types, incomplete statistics show that the length of OSM roads with surface type information in a single country usually accounts for less than 30%, meaning that most OSM road data lack surface type information, highlighting an urgent need for supplementation and improvement." (see **Section 2.2.1**)

(2) Does this study help to address or fill this gap?

Indeed, the purpose of this study is to acquire OSM roads with surface types as training samples. Then, the OSM roads without surface type can be inferred using multi-source geospatial data and deep learning model. Based on the proposed approach, we produced the first road surface type dataset for 50 countries and regions in Africa.

**Reviewer1_Comment6**

In Line 329, the authors selected 5,000 paved and 5,000 unpaved sampling points for each country as training data. However, considering the substantial differences across African countries in terms of geography, infrastructure, and data availability, it would be helpful if the authors could clarify the rationale for using a uniform sampling strategy.

**Reply to Reviewer1_Comment6:** Thanks for this valuable comment! We selected 5,000 paved and 5,000 unpaved sampling points for two reasons:

➢ Firstly, the positive and negative samples are controlled at a 1:1 ratio to achieve equal weights, ensuring sufficient learning for both types.

➢ Secondly, **we found that the model's accuracy improves as the number of sampling points increases**, although it tends to stabilize when the number of sampling points reaches approximately 3,000.

Figure B shows an example of five countries (Cameroon, Djibouti, Egypt, Kenya and Nigeria) that accuracy increases as the number of sampling points increases.

[Figure]

Figure B. An example of five countries (Cameroon, Djibouti, Egypt, Kenya and Nigeria) that accuracy increases as the number of sampling points increases

The above points have been highlighted in the revised manuscript. (see **Section 3.1.3**)

**Reviewer1_Comment7**

In Line 337, the authors provide a brief introduction to the TabNet model. Although the parameter tuning process using the Optuna framework is automated, the optimization objective and models should be described in more detail.

**Reply to Reviewer1_Comment7:** Thanks for this valuable comment! In the revised manuscript, we have added more details about the TabNet model and the Optuna framework. That is,

➢ "TabNet, proposed by Arik et al. (2021), **combines the end-to-end learning and representation learning characteristics of deep neural networks (DNNs) with the interpretability and sparse feature selection advantages of decision tree models.**" (see **Section 3.1.3**)

➢ "The TabNet model is trained, with parameters (e.g., learning rate, number of steps, training epoch) automatically determined using the Optuna framework, which searches for optimal parameters during training. **The core principle of the Optuna framework is to explore various parameter combinations until it identifies the one that yields the highest accuracy. In this study, the search ranges for the parameters—learning rate, number of steps and training epochs—were set to 0.001-0.2, 3-10, and 10-100, respectively.**" (see **Section 3.1.3**)

**Response to Reviewer 2**

**Reviewer2_Comment1**

The calculation methods for the road network features "Degree," "Closeness," and "Betweenness" are not described. Please provide their definitions, formulas, or references to ensure reproducibility.

**Reply to Reviewer2_Comment1:** Thanks for this valuable comment!

The calculation methods for the road network features "Degree," "Closeness," and "Betweenness" are listed as follows:

$$Degree(i) = \sum_{j=1, j \neq i}^{n} k_{ij}$$

$$Betweenness(i) = \sum_{j \neq k \neq i}^{n} n_{jk}(i) / n_{jk}$$

$$Closeness(i) = (n-1) / \sum_{j=1, j \neq i}^{n} d_{ij}$$

Where, $n$ denotes the number of strokes in a road network; $k_{ij}$ denotes the connectivity between stroke $i$ and stroke $j$; $n_{jk}$ denotes the number of shortest paths between stroke $j$ and stroke $k$; $n_{jk}(i)$ denotes the number of shortest paths between the stroke $j$ and the stroke $k$ that contain the stroke $i$; $d_{ij}$ denotes the number of strokes in the shortest path from stroke $i$ to stroke $j$.

Because these "road network features" have been widely introduced in the previous studies (Zhou and Li 2015; Zhou et al., 2025b) for the purpose of street network analysis, thus in the revised manuscript, we added a sentence to highlight this point rather than introduce them again. That is,

"These metrics (road length, Degree, Closeness, Betweenness) are calculated for each stroke, by referring to Zhou and Li (2015); Zhou et al. (2025b)" (see **Section 3.1.2**)

References

1. Zhou, Q., Liu, Y., Liu, Z. Mapping National‐Scale Road Surface Types Using Multisource Open Data and Deep Learning Model. Transaction in GIS, 29(1): 123–141. 2025.

2. Zhou, Q., Li, Z. How many samples are needed? An investigation of binary logistic regression for selective omission in a road network, Cartography and Geographic Information Science. 1545-0465, 20 Nov, 2015.

**Reviewer2_Comment2**

While Phi_k and SHAP are mentioned for feature selection, no results (e.g., correlation matrices, SHAP summary plots) are presented. Please include key results, either in the main text or supplementary materials, to justify the final selected proxies for different countries.

**Reply to Reviewer2_Comment2:** Thanks for this valuable comment!

We had added a figure (see **Appendix A**) to present the selected proxy indicators for 50 African countries. That is,

**Appendix A**

"This figure shows the selected proxy indicators for 50 African countries. For each country, each value in the grid represents the mean SHAP of the corresponding proxy indicator (e.g., road class). Darker colors indicate higher contributions to the classification results. Empty values mean that the corresponding proxy indicator was not used for model training, because it has a high correlation ($> 0.7$) with at least one other proxy indicator but its mean SHAP is lower."

[Figure]

Figure A1. The selected proxy indicators for 50 African countries.

**Reviewer2_Comment3**

The description of the TabNet model training is insufficient. Please specify the hyperparameter search space used with Optuna (e.g., ranges for learning rate, batch size) and the final model.

**Reply to Reviewer2_Comment3:** Thanks for this valuable comment! In the revised manuscript, we have added more details about the TabNet model and the Optuna framework. That is,

➢ "TabNet, proposed by Arik et al. (2021), **combines the end-to-end learning and representation learning characteristics of deep neural networks (DNNs) with the interpretability and sparse feature selection advantages of decision tree models.**" (see **Section 3.1.3**)

➢ "The TabNet model is trained, with parameters (e.g., learning rate, number of steps, training epoch) automatically determined using the Optuna framework, which searches for optimal parameters during training. **The core principle of the Optuna framework is to explore various parameter combinations until it identifies the one that yields the highest accuracy. In this study, the search ranges for the parameters—learning rate, number of steps and training epochs—were set to 0.001-0.2, 3-10, and 10-100, respectively.**" (see **Section 3.1.3**)

**Reviewer2_Comment4**

The validation samples are randomly selected from model predictions. Please clarify if this sampling strategy considers spatial distribution and road class representation to ensure the 1000 points are representative of the entire road network.

**Reply to Reviewer2_Comment4:** Thanks for this valuable comment! In the revised manuscript, the classification accuracy for each of main road classes has also been given out (see **Figure 8** and **Section 5.1**). This is,

[Figure]

Figure 8. The box plot to show the classification accuracy for each of main road classes for 50 African countries.

"For each country and each road class, 100 sampling points were randomly selected for analysis. As shown, most classification accuracies for these road classes are close to or exceed 80%, with some classes—specifically "Motorway", "Trunk" and "Primary"— achieving accuracies above 95%. These results demonstrate the effectiveness of the road surface type dataset, which is consistent with the finding in Figure 4."

**Reviewer2_Comment5**

The accuracy varies significantly between countries (e.g., 77% in Egypt vs. 96% in Burundi). A systematic analysis of the potential causes for this variation (e.g., OSM data completeness, image quality, socio-economic context) is missing and should be added.

**Reply to Reviewer2_Comment5:** Thanks for this valuable comment! In the revised manuscript, A systematic analysis of the potential causes for this variation has been given out. This is (see **Section 5.1**),

"This is likely because the proposed approach relies heavily on the proxy indicator "Road class" (Appendix A), and thus the proportions of various road classes may influence the quality of the developed dataset.

In order to verify this, Figure 8 shows the classification accuracies for nine main road classes in the 50 African countries. For each country and each road class, 100 sampling points were randomly selected for analysis. As shown, most classification accuracies for these road classes are close to or exceed 80%, with some classes— specifically "Motorway", "Trunk" and "Primary"—achieving accuracies above 95%. These results demonstrate the effectiveness of the road surface type dataset, which is consistent with the finding in Figure 4. However, the classification accuracies for the four road classes— "Residential", "Service", "Track" and "Unclassified"—are generally lower than those of other road classes. This is probably because high-class roads are predominantly paved and can be easily identified; in contrast, low-class roads

may consist of a mix of paved and unpaved surfaces, making road surface classification more difficult. Moreover, Figure 9 plots the relationship between the proportions of "Residential", "Service", "Track" and "Unclassified" roads in 50 African countries and the surface type classification accuracies for these countries. This figure shows that the proportions of both "Residential" and "Service" roads have a moderate negative correlation (i.e., -0.405 and -0.527, respectively) with the corresponding classification accuracy of each country. This finding confirms that the proportions of certain road classes (e.g., "Residential" and "Service") may affect the quality of the road surface type dataset. For instance, the higher the proportion of "Residential" roads (e.g., 78% for Egypt), the lower the corresponding classification accuracy (e.g., 77% for Egypt)."

[Figure]

Figure 8. The box plot to show the classification accuracy for each of main road classes for 50 African countries.

[Figure]

Figure 9. The correlation between the proportions of four road classes (a. "Residential", b. "Service", c. "Track" and d. "Unclassified") and corresponding classification accuracies for 50 African Countries.

**Reviewer2_Comment6**

The discussion of discrepancies with IRF data, while noted, could be deeper. Please elaborate on the potential implications of these systematic differences (e.g., consistent overestimation of road length) for downstream applications and the relative advantages of your dataset.

**Reply to Reviewer2_Comment6:** Thanks for this valuable comment!

➤ First of all, the IRF data only provided statistical values for road length and paved road length for each country. That means **the IRF data did not indicate the road surface type for each individual road**. So, the IRF data cannot be used for determining which roads should be improved or paved.

➤ In contrast, **our road surface type dataset can provide the road surface type for each individual road.** Therefore, our dataset can not only provide decision-making support for improving local transportation infrastructure (e.g., upgrading unpaved roads to paved roads) but also be an important data source for assessing SDG 9.1. Furthermore, the dataset can be combined with other data (e.g., population and income) to explore the relationship between road surface types and socioeconomic development.

We have highlighted the above points in the revised manuscript. That is (see **Section 5.2**),

"Compared to traditional statistical data such as those from IRF, the first-ever road surface type dataset for 50 African countries and regions developed in this study not only enables the calculation of statistical indicators such as paved road length and road paved rate for each country and region but also facilitates detailed analyses of which roads are paved or unpaved. This provides valuable decision-making support for improving local transportation infrastructure (e.g., upgrading unpaved roads to paved ones). Additionally, road surface types serve as an important data source for assessing SDG 9.1. Therefore, this dataset can also be combined with population and urban builtup area data to analyze the proportion of rural populations within 2 km of paved or unpaved roads in various African countries (Wanjing et al., 2021), to provide data support for evaluating Africa's sustainable development goals. Last but not least, this dataset can be combined with location data of traffic accidents to analyze the relationship between road surface types and traffic accidents (Patrick et al., 2022); with traffic carbon emission data to analyze the relationship between road surface types and environmental impacts (Ling et al., 2024); or with national income data to analyze the relationship between road surface types and socioeconomic development (Anyanwu et al., 2009)."

**Reviewer2_Comment7**

Several figures (e.g., Fig. 3, 4, 5, 7, 8, 9) are referenced but not provided in the preview. Please ensure all figures are included and are clearly explained in the text. High-resolution versions are essential for review.

**Reply to Reviewer2_Comment7:** Thanks for this valuable comment!

All the figures (e.g., Fig. 3, 4, 5, 7, 8, 9) were added in the revised manuscript. **The DPI of each figure is 600, but the DPI may become lower after uploading them into the submission system for review**.

[Figure]

Figure 3. Visualization of road surface type dataset for 50 African countries and regions

(source: Google Maps. 2025, https://www.google.com/maps/ (last access: 2 Jul 2025)).

[Figure]

Figure 4. Accuracy Assessment Results of the Road Surface Type Dataset.

[Figure]

Figure 5. The Correlation Analysis Results with IRF Statistical Data.

[Figure]

Figure 7. Spatial Pattern Analysis at National, Provincial, and County Levels.

[Figure]

Figure 8. An Example of Road Surface Type Data in Egypt (source: Google Maps. 2025,

https://www.google.com/maps/ (last access: 2 Jul 2025)).

[Figure]

Figure 9. An Example of Explaining the Data Quality of The African Road surface type dataset (source: Google Maps. 2025, https://www.google.com/maps/ (last access: 2 Jul 2025)).

**Reviewer2_Comment8**

The manuscript requires thorough proofreading by a native English speaker or professional editing service to correct grammatical errors and improve sentence fluency (e.g., the phrasing in the abstract "Africa generally have" should be "Africa generally has").

**Reply to Reviewer2_Comment8:** Thanks for this valuable comment!

> First of all, "Africa generally have" has been revised as "Africa generally has" (see **Abstract**).

> Then, the whole manuscript has been polished by a professional editing service company named **Charlesworth** (https://charlesworth.com.cn/author-services.html). Please see the revision in the **Supplement Materials**.

**Reviewer2_Comment9**

The data availability statement is brief. Please enhance it by describing the file format(s), the structure of the attribute table, and providing a direct link or clear instructions for access.

**Reply to Reviewer2_Comment9:** Thanks for this valuable comment!

> We have described the data in **Section 4.1** ("Description of the Africa Road Surface Type Dataset"). This is,

"This dataset was developed based on OpenStreetMap (OSM) road data for Africa, with each country and region stored as a separate vector file in **ESRI Shapefile format**, using **the WGS 1984 Web Mercator projection**. The road data for each country and region include five attribute fields: **road ID, coordinates of the start and end points (see Table 3), road length, and road surface type**. The entire dataset comprises approximately 13,309,000 road segments, with a total length of about 6,822,516 km."

Table 3. Descriptions of dataset

| Attribute | Description | Type |
|---|---|---|
| ID | Road segment ID | Int |
| Start point | Coordinates of the road segment's start point (x, y) | String |
| End point | Coordinates of the road segment's end point (x, y) | String |
| Road length | Length of the road segment (calculated based on WGS 1984 Web Mercator) | Float |
| Surface type | Road surface type, i.e., paved or unpaved | String |

➤ In Section "Data availability", we have highlighted the link for acquiring the data. That is,

"The data can be downloaded from the data repository **Figshare at https://doi.org/10.6084/m9.figshare.29424107** (Liu et al., 2025)."

**Reviewer2_Comment10**

Although the interpretable TabNet model is used, no insights from the model itself (e.g., feature importance rankings for different countries) are discussed. Presenting these findings would significantly strengthen the discussion section.

**Reply to Reviewer2_Comment10:** Thanks for this valuable comment!

➢ First of all, we had added a figure (see **Appendix A**) to present the selected proxy indicators for 50 African countries. That is,

"This figure shows the selected proxy indicators for 50 African countries. For each country, each value in the grid represents the mean SHAP of the corresponding proxy indicator (e.g., road class). Darker colors indicate higher contributions to the classification results. Empty values mean that the corresponding proxy indicator was not used for model training, because it has a high correlation (> 0.7) with at least one other proxy indicator but its mean SHAP is lower."

[Figure]

Figure A1. The selected proxy indicators for 50 African countries.

➤ More important, we have presented the finding in the Discussion (see **Section 5.1**). That is,

"This is likely because the proposed approach relies heavily on the proxy indicator "Road class" (Appendix A), and thus the proportions of various road classes may influence the quality of the developed dataset."

"This finding confirms that the proportions of certain road classes (e.g., "Residential" and "Service") may affect the quality of the road surface type dataset. For instance, the higher the proportion of "Residential" roads (e.g., 78% for Egypt), the lower the corresponding classification accuracy (e.g., 77% for Egypt)."

**Reviewer2_Comment11**

The use of source data from "around 2020" is acknowledged, but the potential impact of minor temporal misalignments between different datasets (e.g., GDP from 2019, population from 2020) on model performance is not discussed. A brief comment on this would be valuable.

**Reply to Reviewer2_Comment11:** Thanks for this valuable comment!

In the revised manuscript, we have highlighted this point in **Section 5.3** ("Limitations and future work"). This is,

"This is because the source data were all obtained from 2020 or nearby years (i.e., 2018 or 2019). **Although existing studies have reported that GDP and building**

**height data change little within a period of 1–2 years (African Development Bank Group, 2020; Ali et al., 2025), inconsistencies in the years may still affect the quality of our dataset**. Therefore, it is worthwhile to investigate whether the quality of the road surface type dataset could be improved by using source data obtained from the same year."

**Reviewer2_Comment12**

The analysis does not break down the classification accuracy by OSM road class (e.g., 'motorway' vs. 'residential'). A stratified accuracy assessment would help identify if performance is biased towards certain road types.

**Reply to Reviewer2_Comment12:** Thanks for this valuable comment!

In the revised manuscript, the classification accuracy for each of main road classes has also been given out (see **Section 5.1**). This is,

"In order to verify this, Figure 8 shows the classification accuracies for nine main road classes in the 50 African countries. For each country and each road class, 100 sampling points were randomly selected for analysis. As shown, most classification accuracies for these road classes are close to or exceed 80%, with some classes— specifically "Motorway", "Trunk" and "Primary"—achieving accuracies above 95%. These results demonstrate the effectiveness of the road surface type dataset, which is consistent with the finding in Figure 4. However, the classification accuracies for the four road classes— "Residential", "Service", "Track" and "Unclassified"—are generally lower than those of other road classes. This is probably because high-class roads are predominantly paved and can be easily identified; in contrast, low-class roads may consist of a mix of paved and unpaved surfaces, making road surface classification more difficult." (see **Section 5.1**)

[Figure]

Figure 8. The box plot to show the classification accuracy for each of main road classes for 50 African countries.

**Reviewer2_Comment13**

The future work section is somewhat generic. Please provide more specific, testable hypotheses or planned methodologies for incorporating terrain data or finer surface type classifications.

**Reply to Reviewer2_Comment13:** Thanks for this valuable comment! We have highlighted this point in the revised manuscript (see Section 5.3). That is,

[revised manuscript text omitted]